# Use of Slaughterhouses as Sentinel Points for Genomic Surveillance of Foot-and-Mouth Disease Virus in Southern Vietnam

**DOI:** 10.3390/v13112203

**Published:** 2021-11-02

**Authors:** Umanga Gunasekara, Miranda R. Bertram, Do H. Dung, Bui H. Hoang, Nguyen T. Phuong, Vo V. Hung, Nguyen V. Long, Phan Q. Minh, Le T. Vu, Pham V. Dong, Andres Perez, Kimberly VanderWaal, Jonathan Arzt

**Affiliations:** 1Veterinary Population Medicine, University of Minnesota, St. Paul, MN 55108, USA; gunas015@umn.edu (U.G.); aperez@umn.edu (A.P.); 2Foreign Animal Disease Research Unit, USDA-ARS, Plum Island Animal Disease Center, Greenport, NY 11957, USA; miranda.bertram@usda.gov; 3Department of Animal Health, Ministry of Agriculture and Rural Development, Hanoi 10000, Vietnam; dung.dah@gmail.com (D.H.D.); long.dahvn@gmail.com (N.V.L.); phanquangminh1@gmail.com (P.Q.M.); dongdah@gmail.com (P.V.D.); 4Regional Animal Health Office No. 6, Department of Animal Health, Ministry of Agriculture and Rural Development, Ho Chi Minh City 71909, Vietnam; huyhoang0778@gmail.com (B.H.H.); nguyenthanhphuong@raho6.gov.vn (N.T.P.); vovanhung.raho6@gmail.com (V.V.H.); trivu78@gmail.com (L.T.V.)

**Keywords:** genetic diversity, phylogenetics, subclinical infection, molecular epidemiology, disease control, surveillance, sentinels

## Abstract

The genetic diversity of foot-and-mouth disease virus (FMDV) poses a challenge to the successful control of the disease, and it is important to identify the emergence of different strains in endemic settings. The objective of this study was to evaluate the sampling of clinically healthy livestock at slaughterhouses as a strategy for genomic FMDV surveillance. Serum samples (*n* = 11,875) and oropharyngeal fluid (OPF) samples (*n* = 5045) were collected from clinically healthy cattle and buffalo on farms in eight provinces in southern and northern Vietnam (2015–2019) to characterize viral diversity. Outbreak sequences were collected between 2009 and 2019. In two slaughterhouses in southern Vietnam, 1200 serum and OPF samples were collected from clinically healthy cattle and buffalo (2017 to 2019) as a pilot study on the use of slaughterhouses as sentinel points in surveillance. FMDV VP1 sequences were analyzed using discriminant principal component analysis and time-scaled phylodynamic trees. Six of seven serotype-O and -A clusters circulating in southern Vietnam between 2017–2019 were detected at least once in slaughterhouses, sometimes pre-dating outbreak sequences associated with the same cluster by 4–6 months. Routine sampling at slaughterhouses may provide a timely and efficient strategy for genomic surveillance to identify circulating and emerging FMDV strains.

## 1. Introduction

Foot-and-mouth disease (FMD) is a contagious disease affecting cloven-hoofed mammals that causes recurrent outbreaks, subclinical infection, and substantial economic losses in affected regions [1]. Foot-and-mouth disease virus (FMDV) is endemic in many developing countries in Asia and Africa, where limited veterinary resources create a need for cost-effective surveillance measures. Surveillance for transboundary animal diseases, such as FMD, typically relies on passive surveillance through outbreak reporting, which sometimes leads to delayed control measures and greater disease spread. Early detection of outbreaks is important to enforce preventive measures and mitigate the impact of the disease, particularly for rapidly evolving RNA viruses, such as FMDV, that have a broad genetic and antigenic diversity. Sampling of animals across the host population to ascertain the prevalence of infection (with or without evidence of clinical signs) is referred to as active surveillance, and can be performed on farms and in animal markets or slaughterhouses to provide a more timely indicator of infection prevalence in a population, particularly if coupled with sequencing to detect emerging variants [2,3,4].

Farm-based active surveillance through randomized sampling would be considered the benchmark of understanding the prevalence and distribution of livestock diseases. Various studies have reported farm-based genomic surveillance of subclinical FMDV strains in endemic regions [5,6,7,8]. However, routine farm-based surveillance is often impractical due to logistical complexity and expense, particularly in rural settings with sub-optimal infrastructure. Slaughterhouses are concentration points where animals from many farms aggregate, and can potentially serve as a convenient, quasi-representative sample of animals from the surrounding host population [9,10,11]. This strategy is employed in veterinary public health to detect diseases or zoonoses of public health concern, such as *Fasciola* or bovine tuberculosis [9,10,12]. Slaughterhouse data, alone and in combination with other variables, have also been utilized for determining the risk factors associated with preserving the quality of meat, and evaluating antibiotic usage in farm animals [13,14]. In most countries, only visual inspections of carcasses are performed in slaughterhouses, though depending on the pathogen, effective disease surveillance can be achieved at slaughterhouses by combining laboratory testing with visual inspection [10,15]. For example, routine slaughterhouse surveillance and laboratory testing to detect emerging diseases is conducted in the European Union (EFSA and ECDC) [16] and the USA (USDA and APHI) [13], though this is not always possible in under-resourced settings with high disease prevalence.

Slaughterhouse-based surveillance is typically passive in nature and is employed for diseases with poor antemortem diagnostic options, and slow-spreading pathogens and parasites that do not require a rapid response; hence it is rarely used for rapidly spreading diseases such as FMD. However, there is substantial and often sub-clinical spread of FMD in endemic countries [1] that is not captured by passive surveillance of reported outbreaks. Active surveillance at slaughterhouses, defined here as the laboratory testing of randomly or purposively selected samples at the slaughterhouse, may provide a cost-effective approach to identifying undetected viral circulation and identifying prevalent or emerging strains. The utility of a slaughterhouse-based genomic surveillance system has not been evaluated for FMDV but could be valuable to improve genomic surveillance in endemic regions for early detection and selection of appropriate vaccines. In addition, monitoring of circulating FMDV strains is a critical component for endemic countries following the Progressive Pathway (PCP) for FMD proposed by FAO/OIE.

Most countries in Southeast Asia (SEA) are FMDV-endemic. In Vietnam, serotypes O and A currently circulate in the country [5]. Serotype O causes 80% of outbreaks, with four distinct lineages present: ME-SA (Mya-98), SEA (PanAsia), O-Ind2001, and Cathay. The PanAsia lineage is currently dominant, having been introduced in 2006 [17]. O/Ind 2001d was introduced into the southern part of the country in 2015 and is currently circulating along with the PanAsia lineage [18]. In addition, the Mya-98 lineage was first identified in Vietnam in 1998 and continues to cause outbreaks and endemic circulation [19]. Serotype A FMDVs identified in the country belong to the Sea/97, genotype IX and are closely related to strains from Laos and Thailand [5,20]. From these observations, it is apparent that FMDV dynamics within Vietnam are characterized by the periodic introduction or emergence of new variants of both serotypes, some of which may become widespread within the country. To develop appropriate control measures or inform vaccine selection, it is important to identify emerging lineages as early as possible. Active surveillance rather than passive outbreak surveillance could provide this opportunity.

The objective of this study was to evaluate active surveillance of clinically healthy ruminant livestock at slaughterhouses as a strategy for genomic surveillance of FMDV under endemic conditions. Specifically, we investigated the extent to which viruses recovered from slaughterhouses reflect the diversity found in the source population (inferred by farm sampling), and whether they can serve as sentinels for the early detection of outbreak strains identified through passive surveillance.

## 2. Materials and Methods

### 2.1. Study Populations and Sampling Design

#### 2.1.1. Farm-Based Sampling

Cattle and buffalo farms from eight provinces in northern (Lang Son, Phu Tho, Bak Kan, Ha Tinh) and southern (Ninh Thuan, Dong Thap, Dak Lak, Binh Phuoc) Vietnam were selected for this study based on their recent outbreak histories and their identifications as FMD hotspots [21,22]. Provinces bordering China (Bak Kan, Lang Son), Laos (Ha Tinh), and Cambodia (Dak Lak, Binh Phouc, Dong Thap) were selected to capture the potential introduction of FMDVs through transboundary movement. The main objective of farm-based sampling was to obtain sequences from circulating viruses. Although we do report the results of the NSP-ELISA and rRT-PCR experiments as part of the sample screening, the intent was not to do a rigorous seroprevalence study. A serial cross-sectional study was carried out across these provinces. Briefly, in each province, 70 to 450 farms (average herd size = 3 animals) were serially sampled between 2015 and 2019, with sampling occurring approximately every 12 months in 2015 and 2016, and every 3 to 4 months over 2017–2019 (Table 1). Sera and oropharyngeal fluid (OPF) were collected from 30 to 250 animals per province per time point (Table 1). Animals that were seropositive against FMDV non-structural proteins (NSP) on the first round of sampling were resampled in consecutive rounds. The number of animals tested from each farm was variable across time, as was the time point in which farms were first initiated into the study. For phylogenetic data analysis, we used an additional 32 sequences obtained from farm-based sampling available from published studies from our group [5,23].

#### 2.1.2. Slaughterhouse-Based Sampling

Two cattle and buffalo slaughterhouses in the Long An and Tay Ninh provinces of southern Vietnam were selected as pilot locations for genomic surveillance (Table 2). These slaughterhouses were selected partly because of their proximity to Cambodia, in order to investigate transboundary movements of FMDVs between these countries, and partly due to animal movement from northern to southern Vietnam [23]. Typically, animals older than 12 months were collected from several farmers in surrounding communes and brought to the slaughterhouses by middlemen. Serial cross-sectional sampling (serum and OPF) was carried out every 15 days from 2017 to 2019. Approximately 30 animals were randomly sampled from each slaughterhouse in each round of sampling.

### 2.2. Outbreak Virus Sequences

Outbreak sequences from across the country were also included in this study to quantify the genetic diversity of FMDV captured by passive surveillance activities. Collection of epithelium and/or OPF from affected cattle, buffalo, or pigs typically occurs when an outbreak (i.e., clinical signs in one or more animals) is reported. However, not all outbreaks are reported, and not all reported outbreaks are sampled. Sampling is usually conducted by the Ministry of Agriculture and Rural Development (MARD), Vietnam, sometimes in collaboration with the United States Department of Agriculture (USDA). In total, 103 and 41 serotype-O and -A outbreak sequences, respectively, were available from 2009 to 2019 from MARD, USDA, and GenBank, which were assumed to represent outbreak samples collected as part of passive surveillance.

### 2.3. Laboratory Analysis

Serum samples were screened for the presence of antibodies against FMDV non-structural proteins (NSP) using a 3ABC ELISA (Priocheck^®^, Prionics, The Netherlands) following manufacturers’ instructions as previously described [5]. OPF and epithelium (outbreak) samples were screened for the presence of FMDV using qRT-PCR [24]. Positive samples were subjected to virus isolation (VI), followed by confirmation of viral RNA in VI supernatant using qRT-PCR, as previously described [24,25]. The VI supernatant RNA was subjected to sequencing using one of several methods, though sequencing was not successful in all cases. Samples from 2013–2015 were sequenced using the Sanger method, as previously described [5], to obtain VP1 sequences, or, by next-generation sequencing (NGS), to obtain full open reading frame (ORF) sequences. For NGS-derived sequences, overlapping RT-PCR amplicons covering the full ORF were produced using three sets of primers [23], and amplicons were sequenced as previously described [26]. Samples from 2016–2017 were sequenced by NGS of RT-PCR amplicons covering the P1 region, as previously described [27]. Finally, sequences from 2018–2019 were sequenced by NGS using random and FMDV-specific primers to obtain the complete genome, as previously described [26,28]. All NGS sequencing was performed using the Illumina NextSeq platform. Read quality filtering, de-novo assembly, and assembly to previously published references of regionally endemic lineages were implemented in CLC Genomics Workbench v12 (Qiagen). Sequences of the VP1 region were utilized in this study. Sequences generated in this study were deposited in GenBank, accession numbers OK205893–OK206077 and OK318499–OK318551.

### 2.4. Analysis of Diagnostic Data

The proportions of anti-NSP antibody positive and rRT-PCR-positive animals were calculated for each province and for each year for farm-based sampling and for each round of slaughterhouse sampling. To determine whether slaughterhouses infections are good indicators of infection prevalence in the surrounding population, we compared apparent seroprevalence of and percent positive for rRT-PCR (OPF sampling) at slaughterhouses and from farms in neighboring provinces during the same time period.

### 2.5. Phylogenetic Analysis

#### 2.5.1. Identification of Circulating Clusters

In order to evaluate the effectiveness of slaughterhouse genomic surveillance, we first classified sequences into genetic clusters of closely related viruses. Delineation of different clusters allowed us to tabulate when and where distinct FMDV variants were detected.

Using the sequence data for the VP1 region of FMDV, we used a discriminant analysis of principle components (DAPC) to find the optimal clustering of sequences that minimized within-cluster genetic variation and maximized between-cluster distance, following Jombart et al.,2010 [29]. The resulting clusters correspond to clades on a phylogenetic tree. The principal components that encapsulated the majority of variability in the genetic data were then used for the discriminatory clustering analysis for both serotypes O and A. Bayesian information criterion (BIC) was used to determine the number of parsimonious clusters. This analysis was performed with the R package *adegenet* [30].

Sequences from each cluster were blasted against the NCBI and WRLFMD prototype lineages to identify the lineage to which each cluster belonged. The clusters were also compared with the currently used vaccine strains in a maximum-likelihood phylogenetic tree. For large clusters identified by DAPC (>10 sequences), the locations and time of appearance of sequences in different parts of Vietnam were mapped using ESRI ArcGIS.

#### 2.5.2. Time-Scaled Phylogenies

In order to identify the emergence of different viral clusters through time and document the timeliness of slaughterhouse surveillance in detecting new clusters, a time-scaled phylogenetic analysis was performed using the Bayesian Evolutionary Analysis Sampling Tree (BEAST v1.10.4) software for both serotype A (134 sequences) and O (221 sequences). For serotype O, a total of 70 sequences from farm-based sampling, 48 sequences from slaughterhouses, and 103 sequences from outbreaks were included in the analysis. For serotype A, 77 sequences from farm-based sampling, 16 sequences from slaughterhouses, and 41 sequences from outbreaks were included. From serotype O sequences, 169 belonged to O/ME-SA/Pan Asia, 34 to O/SEA/Mya98, 6 to O/Cathay and 12 sequences to O/ME-SA/Ind2001d. All the A sequences belonged to Sea/97 lineage. As the farm sampling was longitudinal, in some cases, the same animal was consecutively sampled at different rounds, resulting in nearly identical sequences from the same animal. In such instances, only the first sequence per animal was included. All available outbreak and slaughterhouse sequences were used. Sequences were screened for recombination prior to further analysis using RDP4 software [31] and aligned using MUSCLE algorithm [32]. The best-fit nucleotide substitution model was the HKY model, which was identified through JMODEL test [33].

A relaxed uncorrelated log-normal molecular clock was tested with four different population models (constant, expansion, exponential, and Bayesian Skygrid), with the marginal likelihood of each candidate model compared using path-sampling and stepping-stone estimators (Appendix A) [34]. Each model was run for 200 million iterations on CIPRES [35]. Tracer 1.7.1 was used to assess the conversion of the chains visually and for effective sample sizes of >200 [36]. A relaxed clock coalescent Skygrid model was selected for both serotypes O and A. A maximum clade credibility (MCC) tree was constructed from 10,000 posterior samples of trees (discarding 10% burn-in), and annotated using *ggtree* [37,38]. The time to most common recent ancestor (tMRCA) of each cluster and their 95% highest posterior densities (95%HPD) were obtained from the MCC tree.

## 3. Results

### 3.1. Descriptive Data (Sample Screening)

A total of 11,875 serum samples and 5045 OPF samples from farms were tested via NSP-ELISA and rRT-PCR, respectively, and 115 VP1 sequences were obtained (Table 1). Overall, 42.4% (95%CI: 32.2–52.1%) of serum samples were sero-reactive against FMDV non-structural proteins (i.e., anti-NSP antibody positive), and 8.8% (95%CI: 3.4–15.1%) of OPF were rRT-PCR-positive; 1200 serum samples and 1200 OPF samples were collected from slaughterhouses, and 64 sequences were obtained (Table 2). Across 16 rounds of sampling, 37.3% (95%CI: 32.9–41.7%) of serum samples had a positive anti-NSP antibody response and 10.6% (95%CI: 4.1–16%) were rRT-PCR-positive in the Long An slaughterhouse, whereas 51.8% (95%CI: 47.3–56.4%) of serum samples had a positive anti-NSP antibody response and 16.7% (95%CI: 9.6–24%) were rRT-PCR-positive in the Tay Ninh slaughterhouse. Detailed summaries of diagnostic results by year and province are reported in Appendix A.

The proportion of animals anti-NSP sero-positive in both slaughterhouses had substantial variability across samplings, and confidence intervals were quite wide due to relatively low sample size per time point (Figure 1A). Thus, it was difficult to pinpoint differences between the two slaughterhouses or discern temporal trends. Farm sampling data were available from two provinces (DakLak and Ninh Thuan), located in the same regions as the slaughterhouses, and were sampled at approximately similar time points. In these provinces, on-farm prevalence was similar to that determined in the slaughterhouses, but the confidence intervals were wide (Figure 1B). Amongst anti-NSP antibody positive animals at slaughterhouses (Long An: *n* = 167; TayNinh: *n* = 231), 30.5% (95% CI: 20–38%) and 30.7% (95% CI: 22–40%) were rRT-PCR positive, respectively (Figure 1B). FMDV VP1 sequences were obtained for (179/568) 32% of rRT-PCR-positive OPF samples, reflecting that the acquisition of sequences from OPF samples can be challenging due to low virus load.

### 3.2. Cluster Analysis

For both serotypes, the first nine principal components accounted for 90% of the variability in the genetic data. Through application of DAPC using these nine components, eight clusters were identified based on genetic diversity within serotype O and eight clusters were identified within serotype A. An examination of the number of sequences isolated per cluster through time reveals a pattern whereby a particular cluster emerges (or is first detected), peaks, and subsequently declines in frequency through time (Figure 2A,B). For serotype O, four clusters belonged to the MESA-Pan Asia lineage, two clusters belonged to SEA/Mya-98, one cluster belonged to O/ME-SA/Ind2001d and Cathay lineages clusters, respectively (Figure 3 and Appendix A). For Serotype A, all clusters belonged to the Sea/97 lineage (Figure 4 and Appendix A). Six and four serotypes O and A clusters, respectively, had >10 sequences, each with an average within-cluster genetic distance of 1.0–6.6% in the VP1 region. Appendix A show details of clusters with more than ten sequences, including the lineage to which they belong, place of isolation across years, species, and within- and between-group genetic distances for both serotypes O and A.

Some (17/56, 30.1%) sequences in serotype A-cluster A-9 were previously identified as recombinant sequences within a different study analyzing full-length sequences [39]. Although the VP1 portion of these viruses is not recombinant and belongs to A/Sea-97, other parts of the genome belong to O/ME-SA/PanAsia. Due to the phylogenetic clustering of these 18 sequences with other sequences for which full-length genomes were not available, it is likely that all sequences within this cluster were the same A-O recombinant.

### 3.3. Phylogenetic Data Analysis

To evaluate the utility and timeliness of slaughterhouse surveillance, we focused only on the large clusters (>10 sequences per cluster, Table 3) that were identified in the southern part of the country during the time period in which active sampling was conducted at slaughterhouses in this region (2017–2019). Four and three clusters met these criteria for serotypes O and A, respectively. Of these seven serotype-O-and-A clusters circulating in southern Vietnam at this time, six were detected at slaughterhouses, which suggests that slaughterhouse sampling is effective for revealing the diversity of circulating FMDVs in the host population (Figure 3 and Figure 4, Appendix A). The one cluster which was not detected at slaughterhouses was one that only contained outbreak sequences from pigs (O/Mya-98, Cluster O-6), which were not sampled within as part of farm-based or slaughterhouse surveillance efforts.

For one of the six clusters detected at slaughterhouses (Serotype O cluster O-2), detection through active slaughterhouse surveillance preceded passive outbreak surveillance by 4–6 months (Figure 3). Specifically, the O-2 sequences associated with outbreaks in northern Vietnam in 2018 were detected in slaughterhouses in southern Vietnam in 2017 (Figure 5). The time to the most recent common ancestor for the entire cluster was late 2015 (2015.9, 95%HPD 2013.4–2019.6), and the earliest detection of this cluster in a slaughterhouse was in January 2017 (Table 3). For three clusters in serotype O (clusters O-8, O-9, O-10) and one cluster in serotype A (cluster A-4), clusters were detected in outbreak samples before appearing in active farm and slaughterhouse samples. However, the outbreak samples occurred during time periods during which no active surveillance was conducted for four of these clusters.

## 4. Discussion

This study demonstrates that, in endemic settings, active surveillance of clinically healthy animals at slaughterhouses can potentially be an effective means of genomic surveillance for FMDV. We identified six distinct serotype-O and four serotype-A genetic clusters through sequencing FMDVs recovered from serial cross-sectional sampling at selected slaughterhouses in southern Vietnam, active surveillance at farms, and passive surveillance based on outbreak reporting throughout the country. The data herein indicate that most (six out of seven) large clusters circulating in southern Vietnam between 2017–2019 were detected at least once at slaughterhouses. In addition, our results suggest that, in some cases, slaughterhouse-based surveillance can provide more timely information on circulating or emerging FMDV variants as compared with passive detection through outbreaks. Specifically, some clusters were detected at slaughterhouses four to six months prior to their association with reported outbreaks. These results demonstrate the potential utility of systematic genomic surveillance across a network of slaughterhouses in an endemic country for monitoring circulating FMDV strains, which is a key activity necessary for countries moving through the progressive control pathway (PCP) for FMD proposed by FAO/OIE. Although the current study focused on a relatively small endemic nation, a similar approach could be regionally applied to areas of identified high risk in larger endemic nations.

While slaughterhouse surveillance was able to capture the underlying diversity documented in farms of the same region, proportion positivity for FMDV RNA detection (rRT-PCR) and sero-reactivity (NSP-ELISA) were highly variable through time, which precluded making any conclusions about the representativeness of slaughterhouse samples for estimating prevalence. This was further complicated by the difference in the time schedule of sampling at slaughterhouses and farms, and insufficient sample sizes per time point. Both sampling efforts were not truly random. NSP-ELISA is an antibody test and thus reflects previous exposure at some point in the past, whereas rRT-PCR looks for viral nucleic acid and thus only reflects current (or very recent) infection, which is why the percent positive was higher for ELISA than rRT-PCR. As these slaughterhouses were in border provinces, some animals may have arrived through transboundary animal movements, which may not be representative of seroprevalence in farms in the region. Although slaughterhouse-based sampling may not provide precise estimates of prevalence, routine genomic surveillance at slaughterhouses may be effective for early detection of novel FMDV variants.

Within the scope of this study, circulating viruses in Vietnam were associated with the serotype A Sea/97 lineage and the serotype O/Cathay, O/ME-SA/Pan Asia, O/ME-SA/Ind2001d and O/SEA/Mya-98 lineages, with PanAsia being the most common. This finding is consistent with other recent molecular epidemiology studies in Vietnam [5,23,40]. Although it is present in the region, the O/ME-SA/Ind-2001d lineage only appeared sporadically in Vietnam. According to Vu et al. (2017), this lineage was first detected associated with several outbreaks in 2015, but then subsequently was not detected for 20 months. Analysis of (355) viral sequences collected from slaughterhouses, farms, and outbreaks revealed eight genetic clusters within these lineages. These genetic clusters do not correspond to the spatial clustering of outbreaks reported in different parts of Vietnam [21]. For example, the 90 sequences belonging to serotype O-cluster O-2 were found throughout the country (Figure 5). Viruses isolated from slaughterhouses clustered together with viruses recovered from farms during the same period, indicating that slaughterhouses are representative of FMDV circulation at the farm level. Indeed, six out of seven clusters identified in southern Vietnam from 2017–2019 were detected at least once at these two slaughterhouses. The one cluster not detected in slaughterhouses was comprised exclusively of outbreak samples from pigs, demonstrating a limitation of the active surveillance schemes in this study (sampling ruminants at slaughterhouses misses lineages with tropism for pigs) [19]. Nonetheless, the diversity of FMDVs detected at slaughterhouses was largely representative of the diversity identified in the general population, as quantified from farm-based sampling and passive surveillance.

Sequences identified from Vietnam were closely related to viruses isolated from adjacent countries, indicating a role of transboundary animal movement for FMDV spread and highlighting the importance of regional approach to control FMD in Vietnam [23]. For example, sequences in cluster A-9 were closely related to a serotype A/Sea-97 subgroup-B sequence (A/TAI/1/2012) identified in Thailand [41]. In order to identify and control incursions of novel FMDV variants promptly, it is important to incorporate genomic surveillance as part of routine surveillance at key locations. Our results demonstrate how monitoring slaughterhouses in southern Vietnam could potentially provide early detection of novel variants introduced from neighboring countries. Rather than implementing slaughterhouse surveillance across the entire country, it could be more cost-effective to employ a risk-based approach, whereby a network of sentinel slaughterhouses could be strategically established with consideration to transboundary animal movement and outbreak hotspots. Our results suggest that such a network could identify new FMDV variants in a similar timeframe and in some cases earlier compared to the current status quo of passive surveillance. Such early warning could provide more time for authorities to decide on appropriate control measures and vaccine selection.

Slaughterhouse sampling did not result in earlier detection of genetic clusters in all cases. For clusters that were detected through outbreak sampling (passive surveillance) prior to subclinical detection (active surveillance at slaughterhouses), the outbreak data was not aligned spatially or temporally with the period in which slaughterhouse sampling was conducted. Thus, the apparent delay in detection at slaughterhouses relative to outbreak reporting may reflect that the cluster was not circulating in populations near the slaughterhouses during the period of sampling. However, a larger network of slaughterhouse-based surveillance throughout the country may have detected such clusters earlier.

Time-scaled phylogenies illustrated that closely related viruses were identified in farms both before and after they were detected in association with an outbreak. Animals sampled in slaughterhouses and farms did not have clinical signs of FMD at the time of sampling, and thus detection of virus in such animals represented either persistent infections in carrier animals or early (acute) sub-clinical (neoteric) infections [1]. Related to this, the recovery of viruses in OPF samples collected from persistently infected carriers introduces some uncertainty in the dating of the incidence of infection, as the sample collection date was surely later than the infection date [6,42,43]. This could potentially have impacted the date estimates in the time-scaled phylogenies, though we do not think that it changes our general conclusions about the representativeness and timeliness of slaughterhouse-based surveillance. This study was mainly focused on bovine farms and bovine slaughterhouses and is not representative of the FMD situation in the pig population of Vietnam. However, a similar method would be applicable for pig farms and slaughterhouses in future studies or surveillance programs.

It is apparent from our data that genetic clusters emerged and disappeared over time. Unfortunately, the nature of this study did not allow for the examination of drivers of cluster emergence. As cross-protection amongst related strains may only be partial, immune-driven interactions among co-circulating viruses at the population level could lead to the replacement of existing clusters with new clusters. Cross-protection may result in clinical protection from a different strain of the same serotype, but still may allow for viral replication, transmission, and immune-mediated selection, thus creating ecological or evolutionary selection pressures for viral evolution and cluster turnover. A similar phenomenon of serial subclinical infections with distinct heterologous and homologous strains of FMDV was demonstrated in Asian buffalo in Pakistan [8]. Alternatively, FMDV evolution and circulation of specific genetic clusters in endemic settings may be a product of stochastic spatiotemporal processes (e.g., founder effects) within heterogeneously structured host populations [44], which combine to generate a pattern of introduction, spread, and fade out of clusters over time.

## 5. Conclusions

Active surveillance plays a key role in controlling contagious diseases such as FMD [45,46]. The effectiveness of such surveillance is dependent upon early detection of viral variants using appropriate molecular tools combined with sensibly executed surveillance systems. In this study, we demonstrate a proof-of-concept that active surveillance in sentinel slaughterhouses can capture much of the genetic diversity of circulating endemic FMDVs. Our results suggest that routine genomic surveillance in slaughterhouses would provide representative and timely data on both established and emerging genetic variants, in some cases detecting novel variants four to six months prior to their detection via passive surveillance. These results underscore the potential utility of systematic genomic surveillance for FMDV and other pathogens in slaughterhouses in endemic countries.

## Figures and Tables

**Figure 1 viruses-13-02203-f001:**
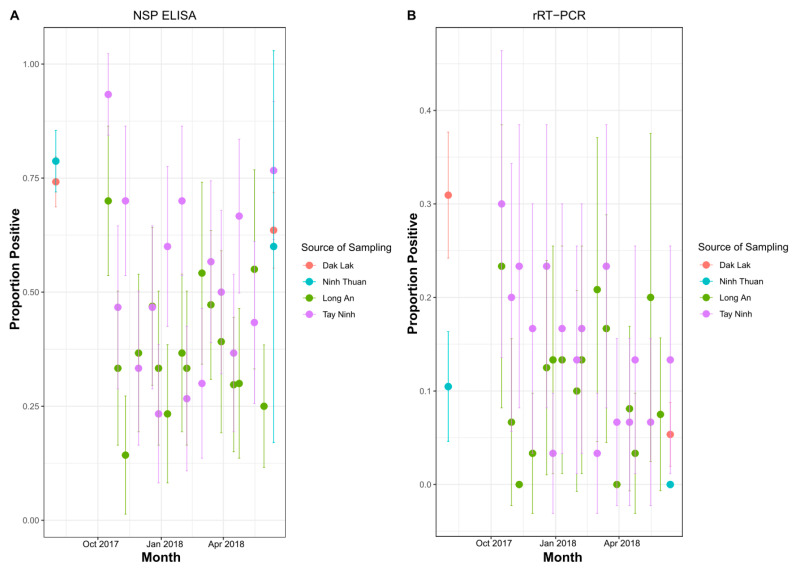
(**A**) Proportion of animals with antibodies against FMD NSPs in farms and slaughterhouses from August 2017 to June 2018. (**B**) rRT-PCR detection rate of FMDV RNA in oropharyngeal fluid from farms and slaughterhouses from August 2017 to June 2018. Error bars represent 95% confidence intervals. Slaughterhouses were in Long An and Tay Ninh. Farms were in Ninh Thuan and Dak Lak.

**Figure 2 viruses-13-02203-f002:**
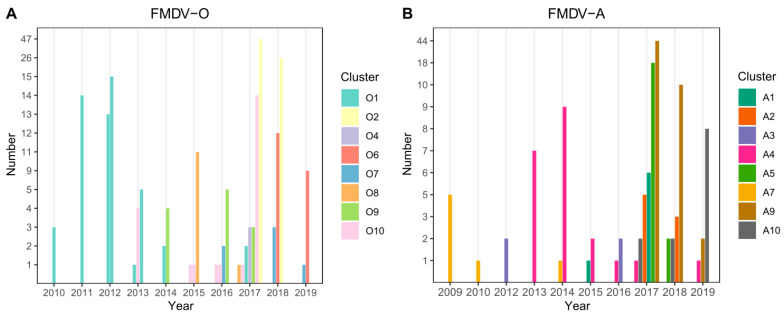
Number of sequences isolated per genetic cluster per year for (**A**) serotype O and (**B**) serotype A from years 2010 through 2019. Serotype O clusters O-6 and O-9 belonged to the SEA/Mya-98 lineage, cluster O-7 to the Cathay lineage, O-8 belonged to ME-SA/Ind2001d and all the other clusters belonged to the O/ME-SA/Pan Asia lineage. All serotype A clusters belonged to the Sea/97 lineage.

**Figure 3 viruses-13-02203-f003:**
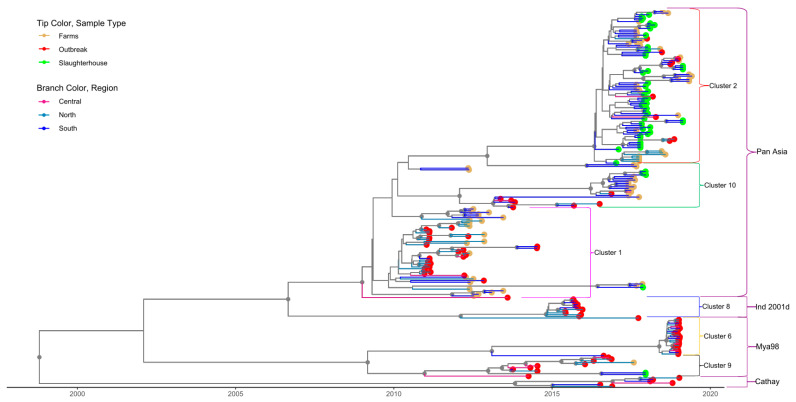
Time-scaled phylogeny for serotype O sequences isolated in Vietnam. Tip color indicates the source type of data (slaughterhouse, farm and outbreak). Different branch colors show the region of the country where sequences were isolated. Small brackets identify the clusters; only clusters with >10 sequences are labelled. The large brackets identify the lineages to which each cluster belongs. Gray circles indicate well-supported clades (posterior probabilities > 0.70).

**Figure 4 viruses-13-02203-f004:**
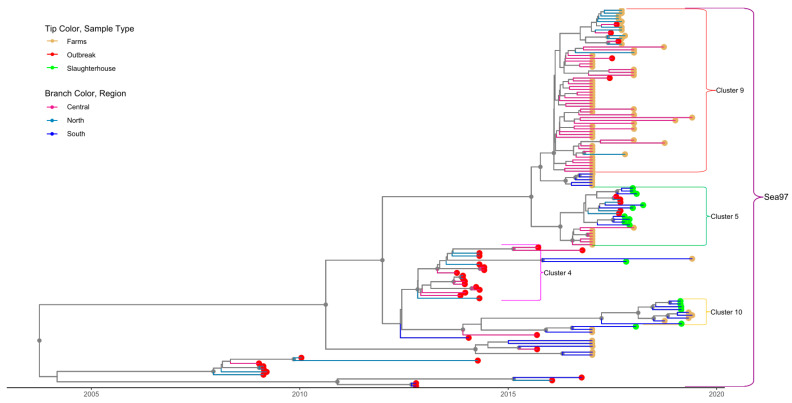
Time-scaled phylogeny for serotype A sequences isolated in Vietnam. All isolates belonged to the SEA-97 lineage. Tip color indicates the source type of data (slaughterhouse, farm and outbreak). Different branch colors show the region of the country where sequences were isolated. Small brackets identify the clusters; only clusters with >10 sequences are labelled. The large brackets identify the lineage to which each cluster belongs. Gray circles indicate well-supported clades (posterior probabilities >0.70.

**Figure 5 viruses-13-02203-f005:**
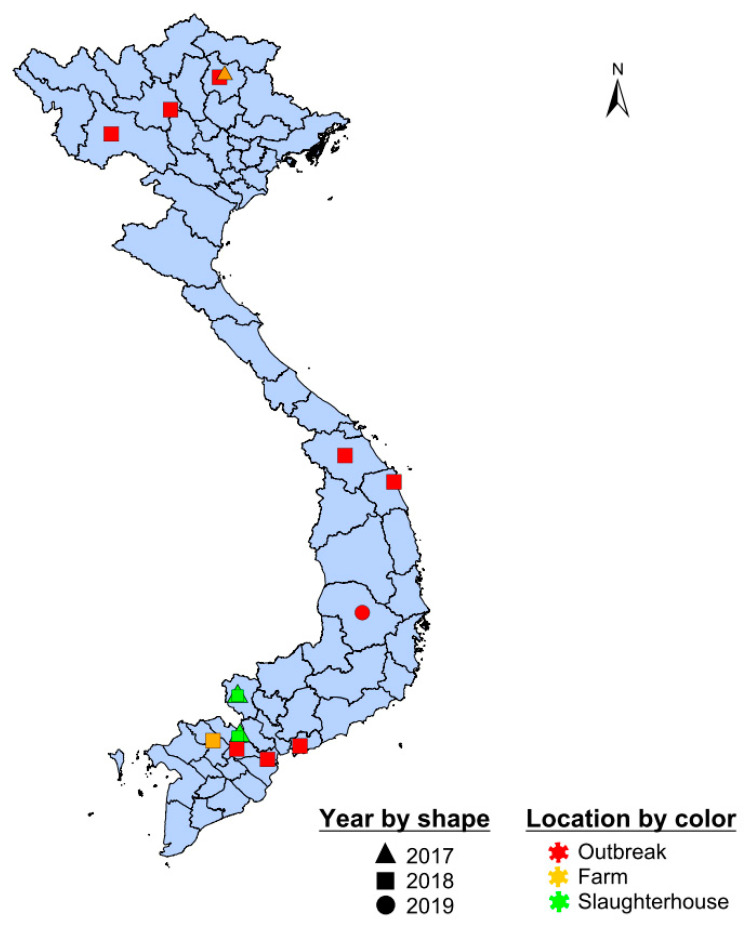
Spatial distribution of sequences in serotype O cluster O-2. Outbreak samples are shown in red, slaughterhouse samples in green, and farms samples in orange. Shape indicates year of sampling.

**Table 1 viruses-13-02203-t001:** Descriptive characterization of longitudinal farm sample screening for FMDV NSP-serology, detection of FMDV RNA in oropharyngeal fluid (OPF), and sequence isolation.

	Province	Sampling Dates	No. of Farms	NSP Serology (Positive/Total); Percent Positive	RNA Detection in OPF Samples (Positive/Total); Percent Positive	No. VP1 Sequences Obtained
Southern Provinces	Ninh Thuan	October 2016 June–September 2017 June–September 2018 January–February 2019	69	(1010/1290); 78.3%	(72/1003); 7.2%	18
	Dong Thap	August 2015 October 2016 June, September–November 2017 June–August 2018 January–February 2019	135	(888/1965); 45.2%	(197/882); 22.3%	30
	Dak Lak	August 2015 August 2017 June–October 2018 January–February 2019	212	(1233/2173); 56.7%	(97/1230); 7.8%	48
	Binh Phuoc	September 2015	160	(84/514); 16.3%	(2/80); 2.5%	0
Northern Provinces	Lang Son	2015 2016 June–September 2017 May–August 2018	227	(208/1387); 15%	(3/223); 1.3%	1
	Phu Tho	2015 2016 August–November 2017 June–September 2018 January–February 2019	442	(269/1256); 21.4%	(2/274); 0.8%	0
	Bak Kan	October 2016 August–November 2017 June–September 2018 January–February 2019	303	(1264/2790); 45.3%	(73/1241); 5.8%	18
	Ha Tinh	August 2015	274	(86/500); 17.2%	(0/112); 0%	0

**Table 2 viruses-13-02203-t002:** Descriptive characterization of slaughterhouse sample screening from two slaughterhouses in southern Vietnam.

Province	Sampling Dates	NSP Serology (Positive/Total); Percent Positive	RNA Detection in OPF Samples (Positive/Total); Percent Positive	No.VP1 Sequences Obtained
Long An	October 2017–May 2018 January–February 2019	(179/480); 37.3%	(51/480); 10.6%	30
Tay Ninh	October 2017–June 2018 January–February 2019	(277/480); 57.7%	(71/480); 14.8%	34

**Table 3 viruses-13-02203-t003:** Summary of clusters with >10 sequences for both serotypes O and A. Sequences were obtained from outbreaks (OB), farms (FA), and slaughterhouses (SH). Regions of the country are divided as northern, central and southern Vietnam. ^†^ Clusters that were circulating in southern Vietnam during period of slaughterhouse sampling.

Serotype/Cluster ID (Lineage)	Source	Number of Sequences per Source	Total Number of Sequences	Region of First Detection	Earliest Date Detected	t MRCA
O-1	OB	26	54			2008.6
(PanAsia)	FA	28		North (FA)	2010-12-22	(1998.7, 2020)
O-2 ^†^	OB	9	90	South (SH)	2017-01-10	2015.9
(PanAsia)	FA	22				(2013.4, 2019.6)
	SH	42				
O-6 †	OB	21	21	South (OB)	2018-02-07	2017.9
(Mya-98)						(2017.5, 2019.1)
O-8	OB	12	12	North (OB)	2015-06-02	2011.7
(Ind2001d)					(2006.6, 2020.9)	
O-9 ^†^	OB	10	13	Central (OB)	2013-10-07	2013.1
(Mya-98)	FA	2				(2007.2, 2019.2)
	SH	1				
O-10 ^†^	OB	2	22	South (OB)	2013-05-17	2009.7
(PanAsia)	FA	3				(2002.9, 2018.8)
	SH	9				
A-4 ^†^	OB	19	21	Central (OB)	2013-10-09	2012.4
(Sea/97)	FA	1				(2006.4, 2019.4)
	SH	1				
A-5 ^†^	FA	6	20	Central (FA)	2017-01-08	2015.8
(Sea/97)	OB	5				
	SH	9				(2013.1, 2019.5)
A-9	FA	5	56	Central (FA)	2017-01-08	2015.3
(Sea/97)	OB	50				(2012.2, 2019.5)
A-10 ^†^	FA	6	12	South (FA)	2018-10-03	2016.8
(Sea/97)	SH	6				(2015.5, 2020.1)

## Data Availability

239 sequences encoding the VP1 region of the FMDV genome that support the findings of this study have been deposited in GenBank (accession numbers OK205893–OK206077 and OK318499–OK318551).

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
