# Peer review of "Use of Slaughterhouses as Sentinel Points for Genomic Surveillance of Foot-and-Mouth Disease Virus in Southern Vietnam"

_viruses, 2021, doi:10.3390/v13112203_

Round 1

Reviewer 1 Report

  1. A high percentage of animals were found positive for FMDV RNA in the OPF samples collected at the slaughterhouses. Therefore, it seems that must of the animals in the slaughterhouses are either in the convalescent phase of FMDV-infection or in the generalized sub-clinical stage of FMDV infection (not FMDV-persistence). Therefore, the authors are requested to provide information on the age of the animals and migration/traffic pattern of animals to the slaughterhouse. Furthermore, whether the sampling period at the slaughterhouses coincides with the field-outbreak?
  2. Data on virus isolation should be provided in the manuscript along with the data of NSP-serology and RT-PCR result.
  3. Although, slaughterhouses-based active surveillance of FMDV would be an useful epidemiological tool for small country, how the strategy may be applicable to large FMD-endemic countries in Africa and Asia?  
  4. Please improve the color pattern of Fig.2. It has been difficult to distinguish between various virus-clusters (especially for serotype A). 

Reviewer 2 Report

Gunasekara et al. analysed FMDV molecular sequences sampled from asymptomatic animals using three different methods of sampling: direct farm samplings, slaughterhouse sampling, and outbreak sampling. The aim of the study is to evaluate sampling of clinically healthy livestock at slaughterhouses as a strategy for genomic FMDV surveillance, which is interesting. Although the authors claimed that their results suggest that “slaughterhouse-based surveillance can provide more timely information on circulating or emerging FMDV variants as compared to passive detection through outbreaks.”, the data and the results presented do not quite support this strong claim. Their analyses and discussion can be improved.

Comments

Line 107: “Animals that were seropositive for FMDV non-structural proteins (NSP) on the first round of sampling were re-sampled in consecutive rounds.”. Were the positive rates reported in Table 1 corrected for this non-independent sampling protocol? If so, how? If not, why would this be acceptable in this circumstance?

Table 1: VP1 sequences were obtained only for a fraction of the samples detected as positive for the virus RNA. Why, and what were the sampling criteria used?

Table 1 and 2: There are clear and consistently large discrepancies between NSP serology positive and RNA positive rates. While these results have been described in text (lines 216-226) and they might not be entirely unexpected, the discussion regarding these results, however, is entirely missing from the manuscript.

Line 204: The results from the model comparisons are not presented in the current manuscript.

Line 218 (and more): The manuscript reads “[o]verall, 42.4% (95%CI: 32.2-52.1%) of serum samples were NSP-positive”. However, the authors actually tested for “the presence of antibodies against FMDV non-structural proteins (NSP) using a 3ABC ELISA” (line 144). “NSP-positive” isn’t the same as positive for antibodies against NSPs. The term is being used inappropriately multiple times, and this could confuse the readers. Please check throughout the manuscript.

Line244: Please discuss the rationale behind performing clustering analysis given that phylogenetic analysis was also performed, and that the two phylogenies (Figure 3 and 4) can provide pretty much all the information that Figure 2 can, if not more. In addition, based on the figures presented, it is clear that a few clusters are not monophyletic, which again is perhaps not unexpected; however, the authors should discuss the discrepancy.

Line 270: The authors stated that “[t]o evaluate the utility and timeliness of slaughterhouse surveillance, we focused only on the large clusters (>10 sequences per cluster)…”. However, I cannot quite see the rationale behind this decision. In fact, the sequence data generated by this study were already highly limited, and I would recommend using all of the sequences available in this analysis. Actually, I would even suggest adding Vietnam FMDV sequences from public databases, like GenBank, to the analysis, and this might improve the results further.

Line 270 & 401: The results from tip-dating analysis are currently being discussed solely in terms of tree topology. I strongly suggest discussing the estimated tMRCAs as well, perhaps by matching them with known past events, for example. In addition, BEAST should have returned effective population size through time to the authors as well. Why not present and discuss the results (against epidemiological data collected through some other means) in the work?

General mood and tone of Discussion: there are a number of instances that the authors overclaim and over discuss their results. For example, the authors write that:

  • This study provides a proof-of-concept that, in endemic settings, active surveillance of asymptomatic animals at slaughterhouses can be an effective means of genomic surveillance for FMDV.” (line 331)

However, the results are way too weak to “prove” this statement. Should the authors wish to claim this statement, more data and statistical analyses are required.

  • “Our results suggest that slaughterhouse-based surveillance can provide more timely information on circulating or emerging FMDV variants as compared to passive detection through outbreaks.” (line 338)

However, no formal statistical comparisons were performed in this study. In fact, the authors than later mentioned that “slaughterhouse sampling did not result in earlier detection of genetic clusters in all cases.” (line 392). The authors could be more precise and should not over discuss their results.

  • This study demonstrates that sentinel surveillance at slaughterhouses is convenient and inexpensive,…” (line 354).

While this might be true, this work does not provide data supporting this discussion.

  • Our results demonstrate how monitoring slaughterhouses in southern Vietnam, bordering Cambodia, was able to provide early detection of novel variants that could potentially have been introduced from neighboring countries.” (line 382).

Again, I cannot seem to find or pin point the results that are being used to support this claim in this manuscript.

Overall, the discuss may need to be rewritten to better reflect the results, and let the data speaks for itself more.

General comment about analysis: given that the main focus of the work is to evaluate if sampling of clinically healthy livestock at slaughterhouses could be an effective strategy for genomic FMDV surveillance or not, crude interpretation of the two trees are not sufficient. I strongly recommend the authors to perform formal statistical analyses to address this question.

Figure 1 A & B: It is unclear why the cluster numbers skip. If the cluster numbering schemes in A and B are correlated, please use the same colours for the same cluster, otherwise, please consider renumbering the clusters.

Figure 1 & 2: The texts are too small to read.

Figure 3 & 4: Figures are vertically stretched and the quality is poor. Divergent dates’ HPDs are missing. Clade support values are missing. Also,  I am guessing that the authors only label clusters with >10 sequences on the trees. However, since they write in the manuscript that:

[f]or serotype O, seven clusters belonged to the MESA-PanAsia lineage and two of the clusters belonged to Mya-98 and Cathay lineages. For Serotype A, all clusters belonged to the SEA/97 lineage (Figure 3 and Supplementary Figure S2)” (line 248-251),

the absence of cluster labelling could be very confusing. Please consider remaking the figures and revising the legends.

Figure 5: The figure is of a very low quality, and please consider presenting the figure much earlier in the manuscript, perhaps under the sampling design - method section.

Figure S1, and S2: Again, figures are poorly prepared, and the legends are poorly written.

Minor comments

There are double spacings and missing spacings throughout the manuscript. Words and phrases used are sometimes unusual, for example “[i]n order to document the effectiveness of slaughterhouse surveillance as a vehicle for genomic surveillance….” (line 171). Please check.

Line 250: “For Serotype A, all clusters belonged to the SEA/97 lineage (Figure 3 and Supplementary Figure S2).” -> Referring to the wrong figure.

Reviewer 3 Report

Manuscript ID: viruses-1371979

Manuscript Title: Use of slaughterhouses as sentinel points for genomic surveillance of subclinical foot-and-mouth disease virus in Vietnam

Introduction:

The authors must clarify if they consider slaughterhouse surveillance is active or passive (paragraph 1, 3, 5).

Line 59-62: The concepts mentioned are well established in these countries/regions with lot of investment and deployment of human and technical resources.  But developing and under-developed countries in Asia and Africa where the disease burden is heavy are under resourced both financially and in human resources.  Most countries are in stage 1-3 of the FMD-PCP and very few countries are in stage 4.  The authors must consider these while writing introductions.

Line 88-93: How can clinically normal ruminant be considered subclinical?  The presence of genome in probang sample is post infection and during convalescence.  They can harbour multiple serotypes and genotypes/lineages from past infections.  So the objective of this study is confusing while considering the basic question raised above. 

Results obtained from probang samples collected from animals brought to slaughterhouse are indicative of past infection.  Since the manuscript does not describe any other samples we cannot assume that this study is a proof-of-concept for active surveillance.

Materials and Methods:

Section 2.11 – Farm based sampling:  Not all farms are spread across the three slaughterhouse sampling sites.  These farms are close to the Cambodian border and in Southern Vietnam.  Therefore, they cannot be a true representation for the whole of Vietnam.  Why no farms in Northern and Central Vietnam selected?

Section 2.1.2 – Slaughterhouse-based sampling: What does each round of sampling mean? What is the time frame for each round?

Section 2.2 – Outbreak virus sequences: The paragraph is verbose and so simplify the paragraph.  What do these outbreak sequences represent, how were they obtained and from where and what type of samples.

Section 2.3 – Laboratory analysis: One basic question, did you compare VP1 sequences (meaning amino acid sequences) or VP1 coding nucleotide sequences (1D region of the genome)?  They are different and not the same.

What about antibody ELISA or VNT results for antibodies against structural proteins?  Did you find more than one serotype?

 Section 2.5.1: Lines 175-182 – Need to provide some basic context about this procedure in the introduction as to why this method is different from the existing methods for comparing nucleotide sequences.  A new reader who is not aware of this method will not understand.

Line 175-178: Sentence is not completed; following what?

Line 178-180: Must be in the results section.

Results:

Section 3.1 – Descriptive data (sample screening)

Lines 232-234: Was there a history of FMD in these farms prior to the study, if yes, how long ago?  Were the animals in the farms showing any subclinical infection (nasal swabs)?

Table 1: provide a column to show the year wise sampling details (you had mentioned something like time frame and round in the methods). Also provide the results for each year.  This will indicate if the sample size was adequate to make a hypothesis or a conclusion based on the evidence.

Section 3.2 – Cluster analysis: The samples have indicated presence of O/ME-SA/PanAsia and O/SEA/Mya-98 lineages along with A/Asia/Sea-97 lineage.  The region as a whole (Pool 1 countries in South East Asia) had O/ME-SA/Ind-2001 lineage outbreaks since 2015 with multiple introductions.  You study shows no evidence for that.  Why? 

Thailand has recorded a variant strain of A/Asia/Sea-97 lineage (A/TAI/2012 LopBuri sub-lineage;  Boonsuya Seeyo et al 2020 Virus Research 290: 198166; https://doi.org/10.1016/j.virusres.2020.198166) in circulation.  Did your studies identify viruses with sequences closely related to this new sub-lineage?

Lines 267-269: Are these probang samples? How were they sequenced?  Sanger or NGS?  It could be past infections with two different serotypes? Or otherwise, you are speculating that it could be A-) recombinant?

Table 3: Provide an additional column to specify how many belonged to the different genotype/lineages that are in circulation.

Figures 3 and 4: The colours are not adequately contrasting.  It is difficult to interpret the dendrogram.  Does the tree contain reference sequences for the different genotypes and lineages?

Discussion:

Lines 331-332: The animals in slaughterhouses are mentioned as asymptomatic?  Asymptomatic for what?  If it were for FMD (obviously) they could have been previously infected or may be in convalescent period or could be very young animals?  There is no age description provided anywhere in the manuscript.  How do we judge?

Line 333-335: Did the clusters group based on the year of sampling?

Line 338-344:  This clearly shows lack of surveillance in the endemic setting of Vietnam and this could  be due to lack of resources.

Line 345-348: The results could be a snapshot of the time of sampling rather than the endemic setting.  The samples were not collected throughout the year but only on certain periods.

351-357: When the slaughterhouses are established close to the borders of a neighbouring country, which is also endemic then this is what you expect.  However, we cannot generalise the statement for the rest of Vietnam as no samples were collected from slaughterhouses/Farms in the Northern or Central Vietnam.

Reviewer 4 Report

Dear Authors,

Please find below my comments and suggestion concerning your manuscript reporting on the “Use of slaughterhouses as sentinel points for genomic surveillance of subclinical foot-and-mouth disease virus  in Vietnam.” This proof of concept is of great interest for genomic surveillance of FMD and thus a better control of the disease. A colossal work was performed under endemic conditions and conclusions are well supported by the results, explained furthermore by very clear and well designed figures. I have only one suggestion/question and one minor comment.

Comment/question:

In the section “discussion”, I suggest to discuss further the cost-effectiveness of the sentinel surveillance at slaughterhouses also addressing the issue of its reduced effectiveness in monitoring swine strains.

Minor comments :

Figure 1: this figure would gain to be, if possible, larger, sharper and more contrasted

Round 2

Reviewer 2 Report

The authors have addressed most of my concerns appropriately. In particular, I really appreciate that the authors have now soften the way they discuss the results substantially. The writing has also significantly improved. There are a couple more minor comments that I hope could be easily addressed.

Comments

  1. The authors replied to my initial comment regarding non-monophyletic clusters that “[w]e believe that the reviewer’s observation of some of the clusters not being monophyletic is due to some imprecision in the brackets used in the figures to mark different clusters, wherein the brackets were not perfectly aligned to the monophyletic clades they were supposed to denote, [and] we have corrected this in the new figures.”. They also stated in the revised manuscript that “[r]esulting clusters correspond to clades on a phylogenetic tree” (Line 194) and that “[w]e identified six distinct serotype O and four serotype A genetic clusters through sequencing FMDVs” (Line 338). However, upon re-examination of the revised figures, it is still clear that Cluster O-1 and O-10, and perhaps also O-9, are not clades / monophyletic. Cluster O-2 are not phylogenetic distinct, but actually descended, from O-10, for example. As I previously commented, this kind of discrepancy is not unsurprising, and in fact rather expected, given the phenetic nature of the clustering method and that the sequences were longitudinally sampled over a substantially long period of time (i.e. it is possible that sister taxa might be more genetically dissimilar than non-sister taxa). There isn’t anything wrong about this, but please describe and discuss the results regarding virus grouping more precisely.

  1. The authors discussed their results that “[v]iruses isolated from slaughterhouses clustered together with viruses recovered from farms during the same period, indicating that slaughterhouses are representative of FMDV circulation at the farm level” (Lines 383-385) and that “the diversity of FMDVs detected at slaughterhouses was largely representative of the diversity identified in the general population” (Lines 390-392). They then concluded their paper with that “[o]ur results suggest that routine genomic surveillance in slaughterhouses would provide representative and timely data on both established and emerging genetic variants..” (Lines 456-458). While it is true that viruses from slaughterhouses and farms cluster together phylogenetically, this doesn’t immediately imply that the former is representative of the latter in terms of genetic diversity (although it might be true). Please consider discussing the results more conservatively, otherwise the authors may support their claim using some tests to show that viral genetic diversities in the two settings are not statistically different over space and time while accounting for phylogenetic relatedness.

  1. It is true that all of the revised figures are of high quality now. Nonetheless, graphs / illustrations are still clearly very big in comparison to the texts in the figures. It is quite likely that when they are presented in actual paper, the texts would still be illegible.

  1. The first references of Figure 3 and 4 (Lines 272-273) come before Figure 2 (Line 280).